# In-Line Monitoring and Control of Rheological Properties through Data-Driven Ultrasound Soft-Sensors

**DOI:** 10.3390/s19225009

**Published:** 2019-11-16

**Authors:** Stefania Tronci, Paul Van Neer, Erwin Giling, Uilke Stelwagen, Daniele Piras, Roberto Mei, Francesc Corominas, Massimiliano Grosso

**Affiliations:** 1Dipartimento di Ingegneria Meccanica, Chimica e dei Materiali, Università degli Studi di Cagliari, 09123 Cagliari, Italy; inforobertomei@gmail.com; 2Department of Acoustics and Sonar, TNO, 2597 AK Den Haag, The Netherlands; paul.vanneer@tno.nl; 3Department of Sustainable Process and Energy Systems, TNO, 2628 CA Delft, The Netherlands; erwin.giling@tno.nl; 4Department of Sustainable Transport & Logistics, TNO, 2595 DA Den Haag, The Netherlands; uilke.stelwagen@tno.nl; 5Department of Optomechatronics, TNO, 2628 CK Delft, The Netherlands; daniele.piras@tno.nl; 6Procter & Gamble Eurocor N.V., 1853 Strombeek-Bever, Belgium; corominas.f@pg.com

**Keywords:** non-Newtonian fluid, ultrasound sensor, data-driven, neural network, hybrid approach, viscosity curve, decision support, Industry 4.0

## Abstract

The use of continuous processing is replacing batch modes because of their capabilities to address issues of agility, flexibility, cost, and robustness. Continuous processes can be operated at more extreme conditions, resulting in higher speed and efficiency. The issue when using a continuous process is to maintain the satisfaction of quality indices even in the presence of perturbations. For this reason, it is important to evaluate in-line key performance indicators. Rheology is a critical parameter when dealing with the production of complex fluids obtained by mixing and filling. In this work, a tomographic ultrasonic velocity meter is applied to obtain the rheological curve of a non-Newtonian fluid. Raw ultrasound signals are processed using a data-driven approach based on principal component analysis (PCA) and feedforward neural networks (FNN). The obtained sensor has been associated with a data-driven decision support system for conducting the process.

## 1. Introduction

The competitiveness of the chemical industry depends on its ability to deliver high quality and high-value products at competitive prices in a sustainable fashion and to adapt quickly to changing customer needs. The use of continuous processes is a promising strategy for this goal. Compared to traditional batch processes, continuous production can lead to better product uniformity and can significantly reduce the consumption of raw materials and energy [1]. On the other hand, in continuous flow processes, identifying the cause of specific deviations during production is more difficult when the product continuously flows through the different unit operations. This implies that the effects of process fluctuations can significantly affect the performance of the process [2], with a significant risk of producing large amounts of off-spec product that needs rework or even must be treated as waste, thus causing high cost and high consumption of energy and of raw materials. For the previous considerations, the use of continuous in-line measurements and control of the product quality are mandatory.

The main obstacle to process monitoring and control is the availability of a suitable sensor [3]. Production of non-Newtonian fluids is one of the fields where commercial sensors capable of providing real-time information about the rheological properties and velocity profile measurements are not still available for industrial applications [4]. Rheological properties are obtained using rheometers, which are generally off-line laboratory devices. Current online viscosity measurements require continuous maintenance, they are generally invasive, they may interfere with the layout of the process, and they only provide rheology information on the liquid near the surface of the device protruding into the liquid. Many recent works focus their efforts on non-invasive techniques for measuring the velocity profile or rheology curve in fluids. It is worth mentioning magnetohydrodynamics as one of the emerging techniques for applications based on microfluidics [5,6] and it can be used to measure flow rate in electrically conductive solution. Another promising technique is based on ultrasounds, which provide inline, non-invasive tools that exploit the Doppler echo of the signal through the material. Important applications of ultrasound can be found in design and construction of frequency-domain sensors [7,8,9]. Velocity profile measurements were obtained by Kotzé et al. [10], who proposed a method that combines the ultrasonic velocity profiling technique with pressure difference measurements. The technique is non-invasive and it can be used to measure opaque and concentrated suspensions. In-line fluid characterization of a wide range of non-Newtonian and opaque fluids was described in [11]. The proposed system, named Flow-Viz, exploits ultrasound to detect the velocity profile of the flow moving in the pipe, and it is designed for industrial use.

A new concept to measure the viscosity as a function of shear rate by measuring the liquid velocity profile and the pressure drop over the sensor has been recently proposed [12]. The measurement system is basically composed of a number of piezoelectric transducers positioned on the external side of a pipe, around the circumference, and a tomographic algorithm is applied to solve the inverse problem. In this work, a hybrid approach is used to correlate ultrasound data with viscosity for measuring the quality of a complex mixture during a continuous process. A physics-based ultrasonic post-processing algorithm in combination with a principal component analysis is used for data reduction, the result of which is fed into a neural network. This alternative solution to the tomographic approach is aimed to address the issues related to ill-conditioning of the inverse problem, which can lead to unfeasible results in some process conditions, and reduce the computational time required by the tomographic algorithm. The data-driven approach is also applied to develop a support decision tool, which predicts the percentage of different ingredients to obtain the desired target in the rheological properties of the end product. The information from the sensor is used to correct the action if a deviation from the set-point is detected.

## 2. Materials and Experimental Setup

The product considered in the present investigation is a detergent with complex rheological behavior. In a previous work [13] a viscosity vs. shear rate model was shown to describe experimental data obtained off-line. Some representative curves are reported in Figure 1 for different ingredients concentration, in order to show that the rheological properties can significantly vary with process conditions and the relationship between ingredient concentration and viscosity curve is not trivial. More details cannot be reported for confidentiality reasons.

The experimental set-up used to obtain the necessary data for mapping ultrasound signals into rheology properties has been described elsewhere [4,14], and it is here summarized for sake of clarity. The pilot plant was located at the Brussels Innovation Centre in Belgium (BIC) and it was used to mimic the mass production of commercial detergents. It consisted basically of a series of tanks containing the ingredients for the production, and the main pipe. Each ingredient (detergent matrix, additives, structurant) was pumped from the tank to the main pipe through secondary lines and with a given mass flow rate. Each injection point in the main pipe was separated from the others. The different ingredients flowed along the pipe because of a pressure differential and they were mixed up by means of static mixers positioned between inlets. The product rheology is affected by factors such as rheology of single components but also temperature and polydispersity [15]. The scheme of the experimental set-up is reported in Figure 2.

An in-line sensor was located at the end of the mixing process. Details on the non-intrusive in-line tomographic ultrasonic velocity meter can be found in [12,16]. The measurement system was composed of nine piezoelectric transducers positioned on the external side of a pipe, around its circumference and along its axial axis. The measurement concept uses the fact that the acoustic wave propagation and liquid velocity add up in a vectorial sense: an acoustic wave traveling upstream will travel slower than an acoustic wave traveling downstream. By using every possible combination of transmitting and receiving transducers a large number of different acoustic travel paths through the liquid are obtained. This data can be combined using a tomographic algorithm to obtain the liquid velocity profiles. The algorithm chosen in the device of van Neer et al. [12,16] uses the up-/downstream travel time differences, arrival times and acoustic travel path lengths of the compressional waves traveling through the liquid to calculate the velocity profiles. The sensor uses the first wave arrivals. Thus, as the transducer locations are fixed, the acoustic travel path lengths are known a priori. However, the up-/downstream time differences and arrival times have to be extracted from the raw ultrasonic data for each measurement. Each ultrasonic time trace is a complex signal containing contributions from the desired first compressional wave arrival, its multiple reflections, spurious arrivals of undesired wave modes—e.g., Lamb waves propagating in the pipe wall and refracting into the liquid—spurious signals from bubbles and particles present in the signal, and electrical crosstalk. To extract the desired up-/downstream time differences and arrival times a custom post-processing algorithm [12] was created consisting of a series of steps: (1) chirp compression to boost the signal-to-noise-ratio; (2) multiple instances of time windowing to remove electrical crosstalk and to isolate the desired compressional wave arrival; (3) band-pass frequency filtering; (4) cross-correlation and subsequent interpolation to determine the up-/downstream time differences and arrival times; and (5) the removal of spurious echoes due to gas bubbles based on statistical analysis. The ultrasonic postprocessing may be seen as a physics-based data reduction, where the raw ultrasonic dataset of 72 × 6000 points (36 upstream travel paths and 36 downstream travel paths × 6000 time points in each ultrasonic time trace) of each measurement is reduced to a dataset of 36 × 2 points containing the up-/downstream travel time differences and arrival times. The scheme summarizing the steps carried on by the post-processing algorithm is reported in Figure 3.

In the current work the ultrasonic arrival times, ultrasonic up-/downstream time differences, measurements of the pressure drop over the ultrasonic sensor and the temperature were used as input for principal component analysis and subsequently fed into a neural network model. Moreover, off-line rheological measurements were carried out on the samples collected in the final tank with a TA AR-2000 stress-controlled rheometer (TA instruments Inc, New Castle, DE, USA) equipped with a 45 mm plate-plate fixture. In more detail, the viscosity curve was estimated through a logarithmic shear rate ramp experiment in the range of shear rate [0.1÷1100] s^−1^. Thus, in principle, at any batch of the product, it was possible to associate both a set of ultrasound measurements and at the same time a traditional rheological measurement.

## 3. Problem Statement

As reported in [4], during the continuous production of the detergent, it is important to guarantee the specific rheological and physical properties. This can be accomplished by controlling the process, but in-line measurements of the fluid viscosity are required to make the proper correction if necessary. The non-intrusive ultrasound velocity meter used in this work is based on a first-principle approach to solving the inverse problem. In more detail, the raw ultrasound data are processed to obtain the fluid velocity profile [12]. The local shear rate is obtained by taking the negative derivative of the velocity profile with respect to the radius and the local shear stress is calculated from the pressure drop over the sensor. By dividing the local shear stress by the local shear rate, the viscosity as a function of the shear rate is obtained. The liquid velocity derivative combined with limited measurement time and signal-to-noise-ratio leads in practice to rather noisy viscosity curves. A solution is to fit an appropriate rheological model to the velocity profiles to obtain the viscosity curve. The sensor was successfully tested with water and Newtonian fluids. On the other hand, the estimation algorithm has led to not definitive results when dealing with the detergent under investigation because of the very complex structure of the fluid also related to the anisotropies of the included phase [17]. In fact, the material under investigation showed a not trivial rheological behavior with a shear-thinning dependence strongly influenced by process fluctuations such as temperature and compound concentrations. The model best describing the considered liquid is the Carreau model. However, that does not lead to an analytical expression of the velocity profiles, thus a nonlinear inversion to determine the model parameters needs to be applied. Furthermore, it is worth noticing that the acoustic waves are generated on the outside of the pipe, being the sensor non-invasive. This aspect leads to the generation of guided waves in the pipe wall in addition to the desired compressional waves in the liquid, therefore the received signal is a combination of said compressional waves and the guided waves. The complexity caused by the interference of spurious guided waves reduces the allowed discretization used to describe the velocity profiles and the problem may become ill-posed.

It is also important to notice that the characteristic time of the mixing process is quite small meaning that the computational time required to obtain the velocity profile from signal processing should be minimized if feedback control is implemented. Such considerations motivate the use of alternative approaches, and data-driven models could successfully address this issue [18,19,20].

As reported in previous investigations of this process [4,13], the quality of the product under study is determined through its rheological properties, which can be summarized by the viscosity vs. shear rate curve. Rheological properties can be affected by process disturbances (e.g., temperature, quality of the ingredients), and they can be adjusted by varying the ingredients flow rates which have a major impact on the required rheology. Unlike the common control problem, the reference value is not a point value but infinite points lying on a one-dimensional manifold (the viscosity curve). As previously discussed, [14], the most representative points for the rheological behavior of the considered system are the viscosities (η) of the studied compound at shear-rate ($\dot{\gamma }$) values equal to 0.1, 1, 10, 100, and 1000 s^−1^. This means that it is not necessary to measure the whole curve, but it is sufficient to develop an input-output model that correlates the off-line viscosities measured in correspondence of shear rate values equal to 0.1, 1, 10, 100 and 1000 s^−1^ to the raw ultrasound signal.

An experimental campaign was carried out to record viscosity data at different operating conditions. Figure 4 shows the percentage of two ingredients (solvent and structurant) used in the different experimental runs. In total, more than 100 experiments were carried out, as a compromise between the necessity to give enough information on the process to the data-driven model and reduce the cost of the experiments.

## 4. Method

The data collected in the plant were used to train and validate the data-driven model. A feedforward neural network (FNN) with three layers was used to describe the rheological data obtained off-line, with the upstream-downstream time delays, pressure drop, and temperature. For any experiment a 36 × 2 matrix of measured ultrasound variables was collected, therefore, pre-processing is used to increase the interpretability of the data. Preprocessing methods depend on the objective of the study and on the technique used to model data [21,22]. In the present work, the Principal Component Analysis (PCA) has been used [23], that allows representing the variation present in many variables using a smaller number of abstract factors (or principal components: PCs) which are orthogonal to each other and sorted in decreasing order. The number of significant PCs should be in principle equal to the number of factors actually describing the data. The procedure is represented in the scheme in Figure 5.

The number of PCs considered to build the PCA model was capable to describe the 95% of the variance present in the raw signals from the ultrasound sensor. The parameter calibration of the neural model (training) and its validation were accomplished using 82 examples for each output.

## 5. Data-Driven Model

An iterative procedure was applied to obtain the best neural network structure, in terms of number of inputs, number of hidden neurons, activation function used in the processing element [24]. Different structures were compared considering the mean square error calculated on the training and test data sets. Each training was accomplished using 100 different initial conditions randomly changed.

The best neural network model for the considered set of data consisted of eight inputs (seven principal components and pressure drop), three hidden neurons, hyperbolic tangent activation function in the hidden layer and linear activation function in the output layer. A neural network structure with nonlinear activation function in both the hidden and output layer had been also taken into account, but it led to worst results in terms of determination coefficients with respect to the selected configuration. Figure 6 shows the selected neural network structure, reported as view obtained with Matlab R2018b (Mathworks Inc., Natick, Massachusetts, MA, USA). It is worth noting that input and output data had been scaled so that they ranged from 0 and 1. This procedure was necessary because the order magnitude of inputs and outputs could be very different and could wrongly affect parameter estimation.

## 6. Results

### 6.1. Neural Network Model Performance

The viscosity values calculated by the neural network and the off-line measured values are reported in Figure 7 for the training set and in Figure 8 for the test set. As regards the training set, there is a quantitative agreement between experimental and predicted viscosities. A larger mismatch is however evident at the lowest shear rates (0.1 s^−1^). This aspect may be related to the fact that there were only few chords covering the center of the pipe where, although liquid velocity was high, the corresponding radial derivative was low. Thus, regions at low shear rates seem to be not adequately explored by the ultrasound fixture and the effective signal-to-noise-ratio (SNR) may result lower for the low shear rate viscosity estimates. Incidentally, one should remark that the SNR of the chords through the center of the pipe was however also quite low because of their greater lengths. Another possible explanation of the discrepancies at low shear rates could be also due to the poor performance that was appreciated for the offline measurements at $\dot{\gamma }=0.1\;s^{-1}$.

The performance in the train data set was evaluated by means of the determination coefficient statistics which assumes value *R*^2^ = 0.89, as it is defined in Equation (1):(1)$$R^{2}=1-\frac{\sum_{i=1}^{n}\sum_{j=1}^{m}{\left(\eta_{ij,sper}-\eta_{ij,pred}\right)}^{2}}{\sum_{i=1}^{n}\sum_{j=1}^{m}{\left(\eta_{ij,sper}-\bar{\eta }\right)}^{2}}$$
where *n* is the number of samples used for the training and *m* is the number of shear rate values used to approximate the viscosity curve, $\bar{\eta }$ is the average viscosity value of the considered dataset.

The prediction capabilities have been assessed using a different set of data with respect to the training (test set) and results are shown in Figure 8 and it is possible to assert that the neural model is able to adequately describe the qualitative behavior of the system. The performance in the test dataset was evaluated by means of the determination coefficient statistics (Equation (1)), which assumes value *R*^2^ = 0.93. It is possible to conclude that the estimations are promising and can help the operator to control the process, as discussed in the following section.

### 6.2. Data-Driven Model to Support Operator Decision

The possibility to use raw data to obtain rheology information on the product can be exploited to develop an algorithm that supports the decision of the plant operator in order to track the system towards a given target viscosity curve. This issue was addressed by training a neural network model that computed the flow rate values required to obtain the desired end-product rheology. A corrective action can be also calculated by comparing the measured in-line viscosity of the sensor coupled with the neural network with the set-point.

Data obtained at the same experimental conditions of the previous task were used to relate the viscosities at different shear rates (inputs) to the ingredient flow rates (outputs). In more detail, the inputs were chosen as the viscosities measured at shear rates equal to 1, 10, and 1000 s^−1^ and fluid temperature, while the outputs were the percentage of solvent and structurant. The following considerations were done for selecting the points on the viscosity curve: (i) obtain a parsimonious model because of the limited amount of data for this type of problem; (ii) as already discussed, off-line viscosity at low shear rate (0.1 s^−1^) has the highest experimental error with respect to the other points; (iii) at intermediate viscosity value, the shear rate value $\dot{\gamma }=10\;s^{-1}$ showed to be more informative with respect to $\dot{\gamma }=100\;s^{-1}$ with a lower modeling error; (iv) the mismatch between observed and target viscosities at the highest shear rate must be kept as small as possible for the required properties of the product under investigation. Additionally, the temperature has an effect on viscosity, so its knowledge is required for a more precise calculation of the ingredients flow rates.

The effects of the obtained action can be monitored by using the values measured in-line with the ultrasound sensor and the neural network. The flowsheet of the algorithm is reported in Figure 9 and it is described in the following. First, the target vector of the viscosity values at shear rates equal to 1, 10 and 1000 s^−1^ is used as input for the neural model, which is indicated with f(·) in Figure 9. It is worth noting that ${\underline{\eta }}_{sp}$ indicates the value of the viscosity used as input to the neural model and that coincides with the target as the algorithm is initialized. The neural model, using also the measured temperature, calculates the manipulated inputs, that are the flowrate of structurant ($q_{swt}$) and solvent ($q_{sol}$). If the calculated values are different from the actual one, they are modified and then the effects of the variations are detected by measuring the viscosity by means of the hybrid sensor here proposed. If the distance between the target vector and the measured one is less than the tolerance, the process is in control and no further action is required. If the error is greater than the tolerance, the value of ${\underline{\eta }}_{sp}$ is updated by using a simple proportional action and this value is used to calculate the new value of the flow rates.

The algorithm was validated by carrying out an experiment in the real pilot plant located at the Brussels Innovation Center. Table 1 reports the experimental results. The viscosity ‘start’ values are the initial values of rheology properties and the ‘target’ values are the objective to be achieved. In this case, three steps of the operation supported by the neural model were carried out and it is evident that the calculated ingredients flowrates are acting on the continuous mixing and dispersion process, as a means to achieve the target rheology profile. It is worth noting that the viscosity values reported in Table 1 and Figure 10 were validated by off-line measurements, carried out using the rheometer AR-2000 TA stress-controlled equipped with a 45 mm plate-plate fixture.

## 7. Conclusions

This work was focused on monitoring and conduction of the continuous production of a detergent. The main issues were related to the obtainment of reliable and fast in-line measurements of the rheology properties of the complex mixture and the development of a support decision tool. The former aspect was solved by exploiting a data-driven approach, where the raw signal obtained by an ultrasound sensor was processed to calculate some key points of the viscosity curve of the considered fluid. A hybrid approach was used, where a physics-based ultrasonic post-processing algorithm in combination with a principal component analysis was used for data reduction. Then, a neural network algorithm was exploited to correlate the most informative principal components of PCA to some representative points of the viscosity curve of the detergent. The second task was addressed using a support decision tool based again on a neural network. The data-driven model was trained to identify the percentage of the ingredients to obtain the required viscosity properties and the effects of possible model mismatch were taken into account using the information obtained by the in-line sensor. The support decision algorithm and the sensor were validated by performing an experimental run with the pilot plant and the developed procedure showed to be successful in obtaining the target viscosities. Results are promising, showing that the use of a data-driven approach to process ultrasound data can be an effective approach when the rheological properties of the fluid are very complex.

## Figures and Tables

**Figure 1 sensors-19-05009-f001:**
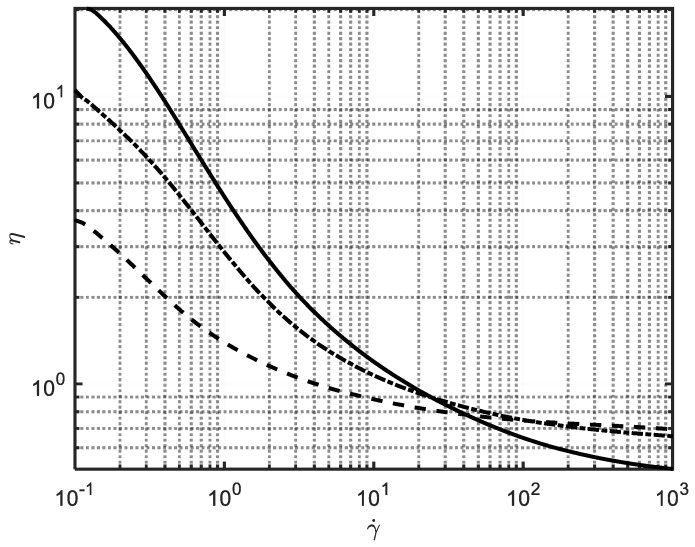
Rheological behavior of the product at different ingredients percentages.

**Figure 2 sensors-19-05009-f002:**
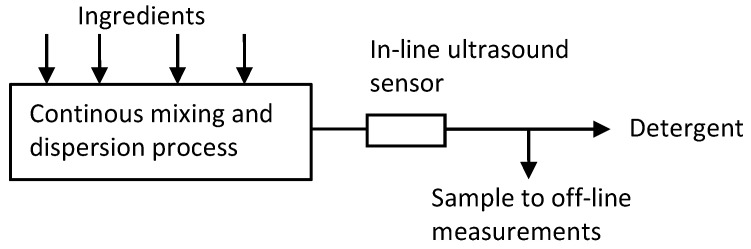
Experimental set-up showing the sensor and the sampling point to collect samples of liquid to bring to the off-line rheometer.

**Figure 3 sensors-19-05009-f003:**
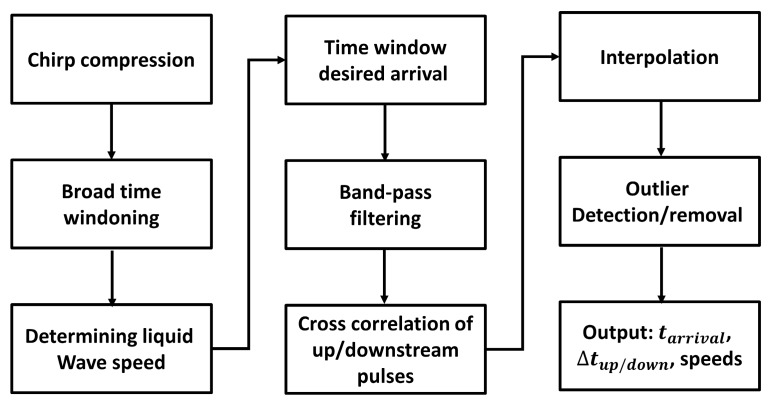
Scheme of the post-processing of ultrasonic signal.

**Figure 4 sensors-19-05009-f004:**
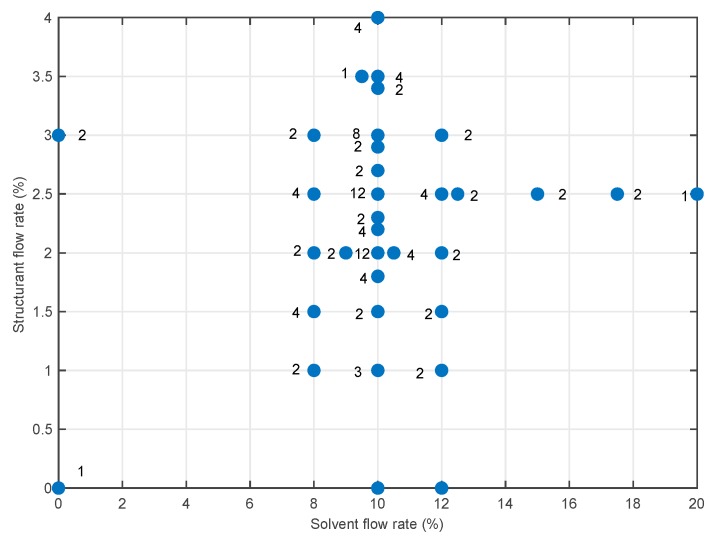
Map of the experiments that have been run in the pilot plant and their composition. The number next to the points is the number of replicates of that setting.

**Figure 5 sensors-19-05009-f005:**
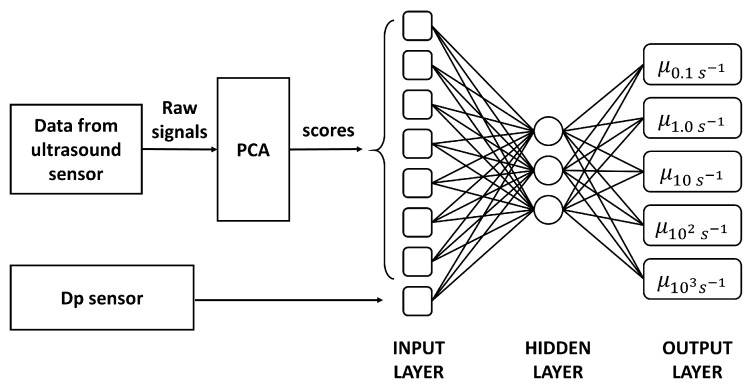
Scheme of the data-driven model for the measurements of rheological properties.

**Figure 6 sensors-19-05009-f006:**
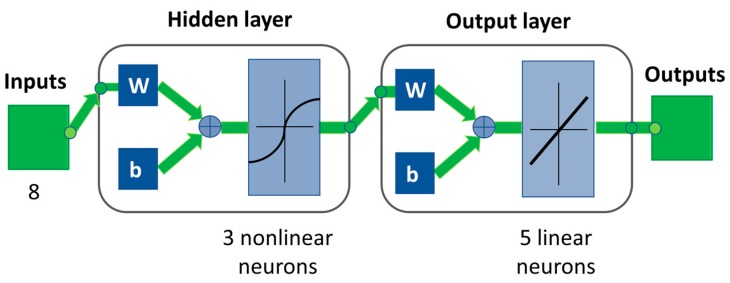
Neural network structure.

**Figure 7 sensors-19-05009-f007:**
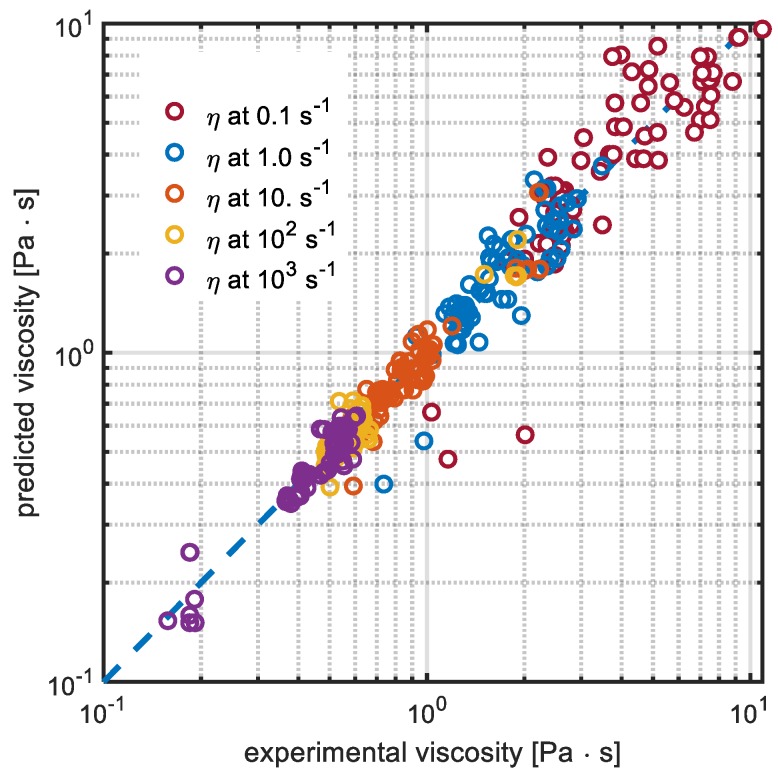
Comparison between experimental and calculated data for viscosity (η) at 0.1 (dark red circles), 1.0 (blue circles), 10 (red circles), 100 (yellow circles), and 1000 s^−1^ (purple circles) for the training dataset.

**Figure 8 sensors-19-05009-f008:**
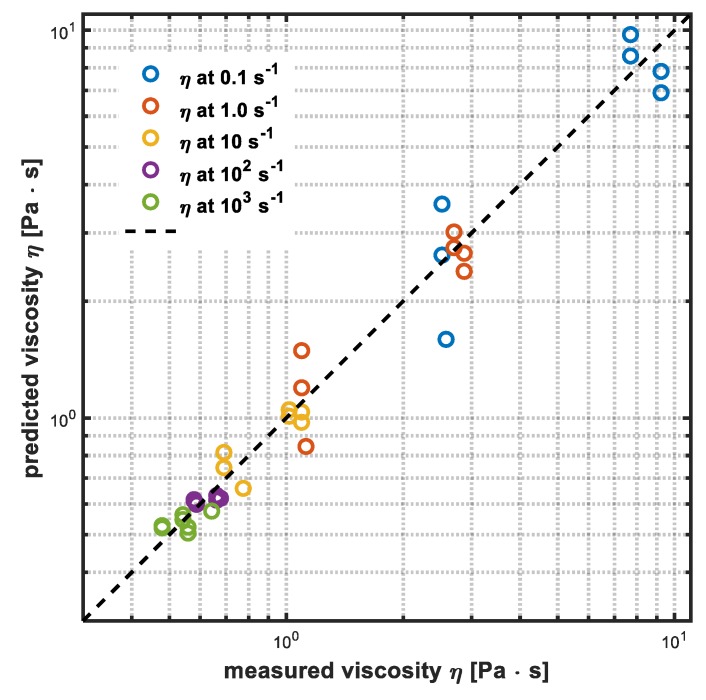
The comparison between experimental and calculated data for viscosity (η) at 0.1 (blue circles), 1.0 (red circles), 10 (yellow circles), 100 (purple circles), and 1000 s^−1^ (green circles) for the test dataset.

**Figure 9 sensors-19-05009-f009:**
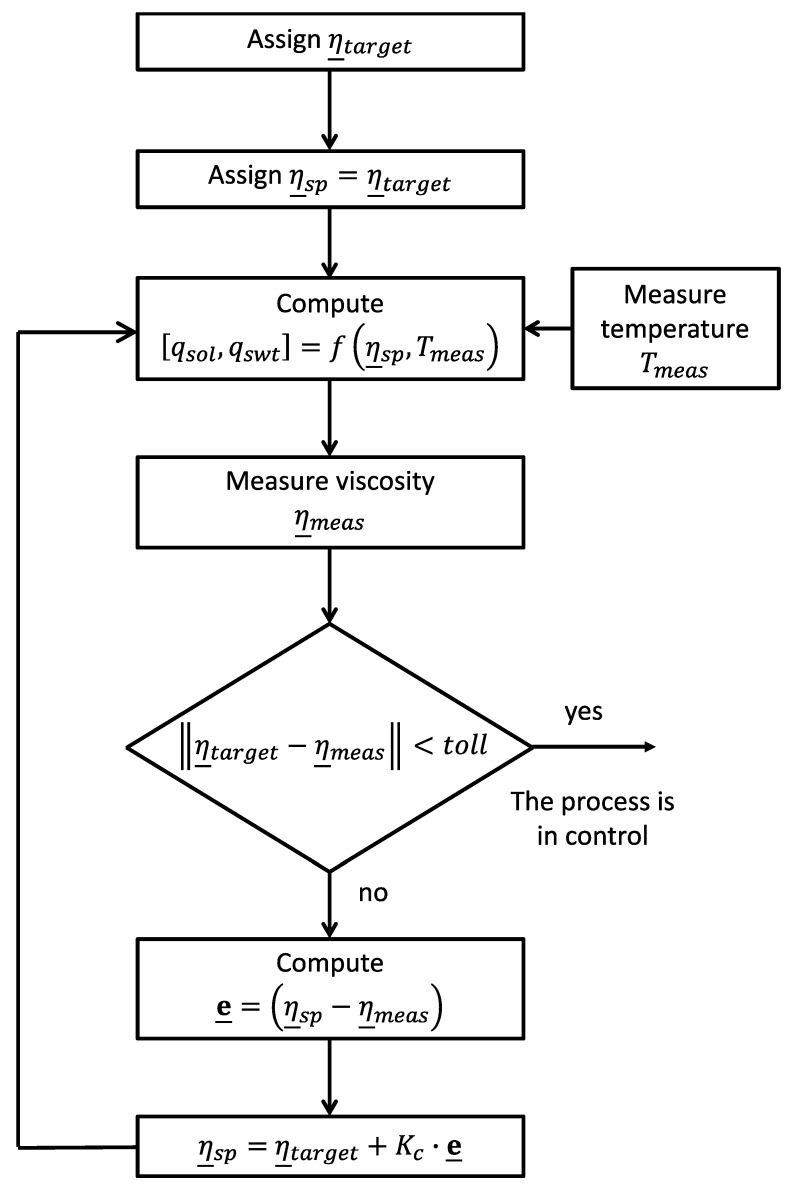
Decision support algorithm.

**Figure 10 sensors-19-05009-f010:**
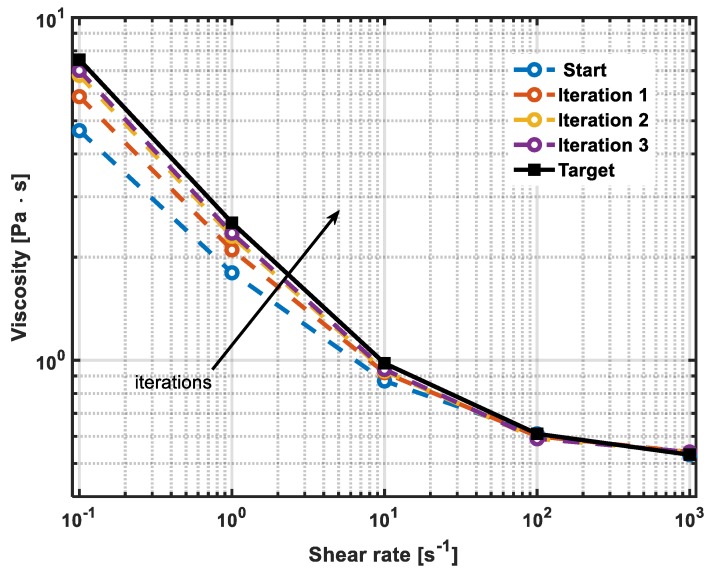
Rheological curves experimentally obtained during the iterations of the control action performed by the decision support algorithm. Measurements are carried out using the rheometer AR-2000 TA stress-controlled equipped with a 45 mm plate-plate fixture.

**Table 1 sensors-19-05009-t001:** Viscosity calculated off-line during the experiment carried out to validate the sensor and the support decision tool.

Shear Rate [1/s]	Viscosity [Pa·s]				
	Start	Step 1	Step 2	Step 3	Target
**0.1**	4.68	5.88	6.78	7.01	7.54
**1**	1.80	2.10	2.30	2.35	2.52
**10**	0.87	0.92	0.93	0.94	0.98
**100**	0.61	0.60	0.59	0.59	0.61
**1000**	0.53	0.54	0.54	0.54	0.53

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
