# Peer review of "In-Line Monitoring and Control of Rheological Properties through Data-Driven Ultrasound Soft-Sensors"

_sensors, 2019, doi:10.3390/s19225009_

Round 1

Reviewer 1 Report

The contribution of the paper is clear. From this reviewer point of view the paper could be published after handling the following issues:

a) The physics based post-processing algorithm is hard to follow and a flowsheet similar to that of figure 7 is suggested.

b) Page 4, line 165, section number is wrong.

c) Besides using figures 5 and 6 to show the data fitting results, a table with mean square error results should be included, in order to quantitatively understand the phrase "assert that the neural model is able to adequately describe the qualitative behavior of the system".

Reviewer 2 Report

After reading the whole document, the abstract yes describe the generalities of what the paper is.

For the introduction dramatic improvement  I suggest:

*Complete the below statement:

The main obstacle to process monitoring and control is the availability of a suitable sensor “and a data-driven decision support system for conducting the process”.

* While talking about Ultrasound techniques and piezoelectric transducers, I consider it is important to mention that they have been used for the design and construction of a big variety of frequency-domain sensors and that novel methods of frequency counting of signals are coming frequently in the state of art to improve their performances. Some references to cite are:

Sergiyenko, O. Y., Balbuena, D. H., Tyrsa, V. V., Mendez, P. L. R., Hernandez, W., Hipolito, J. I. N., ... & Lopez, M. R. (2011). Automotive FDS resolution improvement by using the principle of rational approximation. IEEE Sensors Journal12(5), 1112-1121.

Sanchez-Lopez, J. D. D., Murrieta-Rico, F. N., Petranovskii, V., Antúnez-García, J., Yocupicio-Gaxiola, R. I., Sergiyenko, O., ... & Vazquez-Briseño, M. (2019). Effect of phase in fast frequency measurements for sensors embedded in robotic systems. International Journal of Advanced Robotic Systems16(4), 1729881419869727.

Murrieta-Rico, F. N., Petranovskii, V., Sergiyenko, O. Y., Hernandez-Balbuena, D., & Lindner, L. (2017). A New Approach to Measurement of Frequency Shifts Using the Principle of Rational Approximations. Metrology and Measurement Systems24(1), 45-56.

*It is novelty to use the principal component analysis for data reduction to feed the neuronal network, but it is valuable to comment other options for pre-processing such as outliers eliminations.  Some references to cite are:

Flores-Fuentes, W., Sergiyenko, O., Gonzalez-Navarro, F. F., Rivas-López, M., Rodríguez-Quiñonez, J. C., Hernández-Balbuena, D., ... & Lindner, L. (2016). Multivariate outlier mining and regression feedback for 3D measurement improvement in opto-mechanical system. Optical and Quantum Electronics48(8), 403.

Flores-Fuentes, W., Sergiyenko, O., Gonzalez-Navarro, F. F., Rivas-López, M., Hernandez-Balbuena, D., Rodríguez-Quiñonez, J. C., ... & Lindner, L. (2016). Optoelectronic instrumentation enhancement using data mining feedback for a 3D measurement system. Optical Review23(6), 891-896.

*The introduction is missing to handle the state of art of important concepts:

-Which are the rheological properties of the end product, which is the end product, how has been controlled before?

-Data driven approach by support decision tool based on artificial algorithms, such as Neuronal Networks  and other such as Support vector machine, and others, exist and require to be mentioned. Cite relevant works that used artificial algorithms to obtain decision tool and also used the algorithm  answer to feedback measurement process, such as:

Rodríguez-Quiñonez, J. C., Sergiyenko, O., Flores-Fuentes, W., Rivas-Lopez, M., Hernandez-Balbuena, D., Rascón, R., & Mercorelli, P. (2017). Improve a 3D distance measurement accuracy in stereo vision systems using optimization methods’ approach. Opto-Electronics Review25(1), 24-32.

-It is also important to mention the state of art regarding non-intrusive mixers and how the velocity profile measurements are performed. One example is those based on magnetohidrodynamic, such as:

Valenzuela-Delgado, M., Flores-Fuentes, W., Rivas-López, M., Sergiyenko, O., Lindner, L., Hernández-Balbuena, D., & Rodríguez-Quiñonez, J. (2018). Electrolyte magnetohydrondyamics flow sensing in an open annular channel—a vision system for validation of the mathematical model. Sensors18(6), 1683.

Flores-Fuentes, W., Valenzuela-Delgado, M., Cáceres-Hernández, D., Sergiyenko, O., Bravo-Zanoguera, M. E., Rodríguez-Quiñonez, J. C., ... & Rivas-López, M. (2019). Magnetohydrodynamic velocity profile measurement for microelectromechanical systems micro-robot design. International Journal of Advanced Robotic Systems16(5), 1729881419875611.

*In section 3.

36×2 vector [Should be vector or matrix?]

*In Figures 5 and 6 captions use a different symbol (mu) for viscosity, while inside the figure the symbol (eta) is used for viscosity.

*Table 1 requires more details, start, step 1, step 2 and step 3 were obtained with the proposed tool. But which another measurement system was used to verify that the target really was the value established.

Results in Figure 8 are very promising, congratulations.

Reviewer 3 Report

The paper is interesting and the application is very interesting.

On the other side, too many things keep mysterious.

In fact, one cannot even think of replying results (as there are confidentiality issues).

Then, no comment about the reasons why (for example) it is convenient to use one activation function rather than another one is given. Also the basic equations of the underlying inverse problems are missing (is that a linear or non linear inverse problem ? Why is it ill conditioned ? Is it in principle ill-posed ?)

In conclusion, I think the paper is interesting for the applications, but that more details/discussion is required to come to a nice scientific paper.

And the integration of info/references is part of that (are there alternative non NN based methods ?  why is it ill-posed or ill-conditioned ? Could one use additional sensors ?...)

Reviewer 4 Report

In this paper, a tomographic ultrasonic velocity meter is applied to obtain the rheological curve of a non-Newtonian fluid. Raw ultrasound signals are processed using a data-driven approach based on principal component analysis (PCA) and feedforward neural networks (FNN). The obtained sensor has been associated with a data-driven decision support system for conducting the process. Promising results reveal that the employment of a data-driven approach for processing ultrasound data can be effective when the rheological properties of the fluid are very complex. Fourteen papers were cited in the reference section; however, the number of sources for this paper should be increased. Please revise and enhance the English writing to increase the readability.
